# Cross-Level Influence of Group-Focused Transformational Leadership on Organizational Citizenship Behavior among Chinese Secondary School Teachers

**DOI:** 10.3390/bs13100848

**Published:** 2023-10-16

**Authors:** Zhuotao Fang, Shun-Chi Yu

**Affiliations:** 1College of Educational Science, Minzu Normal University of Xingyi, Xingyi 562400, China; zhuotao.fang@stu.nida.ac.th; 2International College of National Institute of Development Administration, Bangkok 10240, Thailand

**Keywords:** group-focused transformational leadership, psychological contract, organizational citizenship behavior, collectivism, cross-level influence

## Abstract

The organizational citizenship behavior of teachers holds paramount significance in elevating school organizational effectiveness and sustaining competitive advantage. To address this, this study examines the cross-level influence of group-focused transformational leadership on organizational citizenship behavior among secondary school teachers. This exploration includes investigating the mediating role of individual-level psychological contract fulfillment and the moderating impact of group-level collectivism. An empirical investigation involving 1162 secondary school teachers in China was designed for this purpose. The results demonstrate that group-focused transformational leadership significantly positively impacts teachers’ organizational citizenship behavior. Moreover, this relationship is positively moderated by collectivism at the group level, suggesting a stronger effect of transformational leadership on organizational citizenship behavior in more collective-oriented groups. Additionally, the findings reveal that psychological contract fulfillment at the individual level mediates this cross-level relationship, providing evidence for its role in translating leadership’s influence to organizational citizenship behavior. The findings underscore the significance of concentrating on group-focused transformational leadership, cultivating a collectivist atmosphere, and guaranteeing the fulfillment of psychological contracts as pivotal strategies for bolstering organizational citizenship behavior among teachers.

## 1. Introduction

At present, the continuous stream of educational reforms worldwide mandates innovation and flexibility within school organizations to augment their adaptability to environmental shifts and pedagogical transformations. Traditional, fixed job responsibilities are rendered inadequate in the face of such dynamic changes [1,2], hence elevating the significance of teachers’ organizational citizenship behavior. This behavior, metaphorically seen as a “lubricant”, bolsters school organizational efficacy [3].

For example, as the curriculum reform deepens and the emergence of interdisciplinary courses pairs with the growth of research-oriented learning, the roles and responsibilities of teachers should transcend the confines of conventional subject teaching [4]. An essential requisite in this evolving educational scenario is the development of teachers’ awareness and abilities in interdisciplinary teaching, an achievement possible only through extensive collaboration and altruistic assistance among faculty across diverse disciplines [5]. Herein, the organizational citizenship behavior plays a cardinal role, as it encourages teachers to aid others selflessly. This behavior is inclusive of the critical practice of knowledge-sharing in interdisciplinary courses, thereby fostering an environment of collective growth and shared wisdom [6].

Nevertheless, the subtle and non-economic nature of teachers’ work renders many behaviors and performance outcomes challenging to delineate or quantify for assessment and motivation purposes [7]. Hence, if school organizations aspire for teachers to undertake informal tasks that lack comprehensive regulation yet maintain the organization’s competitive edge [8], then the engagement in organizational citizenship behavior becomes of paramount importance for teachers. This non-structured, extra-role behavior can indeed help in advancing the educational mission and promoting an overall beneficial academic environment.

The advent of organizational citizenship behavior can be attributed to many factors, with transformational leadership serving as a notable determinant [9]. Transformational leaders adeptly articulate the organization’s vision to its employees, bolstering employee identification through the presentation of attainable prospects [10]. These leaders inspire employees to exhibit high levels of commitment and altruism, thus encouraging them to prioritize collective interests over individual ones [11]. Moreover, transformational leadership cultivates a congenial organizational atmosphere, facilitating employees in recognizing their self-worth within an engaging working environment and propelling the achievement of organizational objectives [12].

In a recent meta-analytical study, de Geus et al. demonstrated that positive leadership (encompassing both transformational and ethical leadership) exerts a significantly positive influence on subordinates’ organizational citizenship behavior within the public sectors [13]. Despite the abundance of research corroborating the significance of positive transformational leadership behavior for fostering subordinates’ individual-level organizational citizenship behavior [14,15,16,17], the inherent mechanism underpinning how transformational leadership successfully activates subordinates’ organizational citizenship behavior warrants further exploration. Such research will serve to clarify the nuanced interactions between leadership behaviors and their impacts on organizational citizenship actions.

In the realm of transformational leadership, scholars have identified its influence at two distinct levels: the individual and the group level [18]. Group-focused transformational leadership directs its attention towards collective entities, emphasizing shared objectives and interests, fostering group identification, and inspiring effective collaboration towards common goals [10]. Conversely, individual-focused transformational leadership shifts its focus towards individual constituents, tailoring guidance and assistance based on deep understanding, resulting in varied perceptions among group members [10]. While theory suggests that transformational leadership operates at both levels [19,20,21], current scholarly work often treats it as a single construct tied primarily to job performance outcomes [22,23,24]. Scholars advocate for more a nuanced exploration of the impact mechanisms at each level, especially regarding the motivational mechanisms of group-focused and individual-focused transformational leadership [10,19,21,25,26]. Hence, further research is needed to understand the extent to which group-focused transformational leadership influences positive organizational citizenship behavior at the individual level, providing insights into the multi-dimensional effects of leadership styles on individual and group dynamics in various organizational contexts.

In addition, this study addresses another research gap by proposing a mediating mechanism through which group-focused transformational leadership influences the level of organizational citizenship behavior of teachers. The researchers believe that transformational leaders have the potential to enhance followers’ organizational citizenship behavior through various mediating mechanisms and variables [27]. The psychological contract is one of these emphasized mediating variables in the literature [28,29,30]. Past research on psychological contracts has primarily focused on breach [30,31], while, in terms of fulfillment, despite its significance as a predictor of employee attitudes and behaviors [32], it has received limited attention in the existing literature [28]. Therefore, this study proposes the mediating role of psychological contract fulfillment in the relationship between team-oriented transformational leadership and organizational citizenship behavior of teachers.

According to self-determination theory, subordinates may initially take certain actions based on external factors such as encouragement, support, or inhibition from their leaders [33]. However, over time, subordinates may gradually internalize these external stimuli into intrinsic motivations that drive them to engage more autonomously in their work [33]. Within this theoretical framework, group-focused transformational leadership, as a potent situational resource, actively engages in encouraging employee participation in decision-making and sharing common goals [11]. These behaviors (extrinsic motivations) can satisfy subordinates’ basic psychological needs, especially autonomy [34]. As employees progressively recognize the importance of extrinsic motivations, they may internalize these extrinsic motivators and exhibit positive intrinsic motivations in their work [35]. Consequently, they demonstrate positive organizational citizenship behaviors because they perceive these behaviors as voluntary choices rather than being driven by external rewards or pressures [36]. Psychological contract fulfillment may play a mediating role in this process, linking group-focused transformational leadership and organizational citizenship behavior.

Further, this study aims to theoretically elucidate an additional specific moderating mechanism—collectivism. Collectivism, characterized as a sociocultural orientation, privileges collective interests and overall concerns over individual interests [37]. In collectivist cultures, individual behaviors and decisions are typically influenced and guided by the group, family, or organization, predicated on the widely held belief that collective interests supersede individual ones [37]. These cultures prioritize communal values and obligations, emphasize reciprocal aid and cooperative relationships, strive to maintain harmonious and unified social relations, and place high importance on group identity and belonging [38].

Teachers cultivate their professional identity within a framework of collectivist values, adopting these as an integral part of their socialization process. They align their understanding of the teacher’s role, delineating the boundaries of formal and informal behaviors within their personal roles when considering student interests, based on these values [39]. Consequently, they exhibit a greater predisposition towards making sacrifices for the collective good [40].

Some existing empirical evidence points out the moderating role of collectivism in the relationship between transformational leadership and organizational citizenship behavior [41,42]. It can be speculated that one of the mechanisms through which group-focused transformational leadership motivates subordinates is by fostering a collective identity and shared group goals. In groups imbued with a potent collectivist ethos, members are more likely to become homogenized [43]. Group goals are more readily internalized into individual objectives, and group members exhibit a higher likelihood of adhering to rules and trusting their leaders [44]. In essence, within the cultural framework of collectivism, group-focused transformational leadership facilitates easier acceptance and cultivation of group goals, promotes harmony within the group, and simplifies the establishment of trust relationships between leaders and members, thus augmenting leadership effectiveness (Schaubroeck, et al. [45]). By considering the effect of collectivism at the group level, this study enriches the leadership literature, addressing the call for a more nuanced exploration of background characteristics underpinning transformational leadership theory [46,47,48].

This study aims to explore the significance of organizational citizenship behavior among teachers in the context of continuous educational reforms and evolving roles in schools. It delves into how the non-economic, extra-role behavior of teachers contributes to school organizational efficacy and student development. Additionally, this study seeks to understand the role of transformational leadership, at the group level, in motivating organizational citizenship behavior among teachers. Furthermore, it aims to investigate the moderating role of collectivism within the cultural framework in shaping the relationship between transformational leadership and organizational citizenship behavior among teachers. This research work seeks to provide insights into the nuanced mechanisms underpinning the impact of leadership styles on individual and group outcomes, ultimately contributing to more effective leadership practices in diverse educational settings.

## 2. Theoretical Framework and Literature Review

### 2.1. Theoretical Framework

Self-determination theory offers a robust framework for comprehending the motivation and quality of human behavior [49]. The core tenet of this theory asserts that the influence of various environmental factors (including job design, contingency pay, leadership style) on subordinates’ motivation and experience is largely mediated by a set of basic psychological needs (competence, autonomy, and relatedness) [50].

Burns first introduced the concept of transformational leadership, shifting the focus of leadership research towards how leaders can change and transform the values, interpersonal relationships, organizational culture, and behavior patterns within an organization [51]. He viewed leadership as an evolving process of mutual influence between leaders and followers, where leaders and followers work together to stimulate intellect and inspire spirits to drive organizational transformation [51].

Transformational leadership involves leaders making their subordinates aware of the meaning of the tasks they undertake, inspiring high-level needs in their subordinates, and establishing a mutual trust atmosphere [52]. In certain situations, subordinates may even sacrifice personal interests for the benefit of the organization and achieve results that exceed their initial expectations [17,41,53]. This leadership style triggers intrinsic motivation in subordinates, making them more committed to their work, and hence driving organizational change and progress [34].

Kovjanic et al. contend that, viewed through the lens of self-determination theory, transformational leadership can be interpreted as bolstering the quality and quantity of followers’ job performance by supporting their fundamental psychological needs [34]. When these basic psychological needs are sustained by external environmental stimuli, followers perceive these as symbolic and expressive facets of their self-concept and adopt shared values and goals as their guiding principles [50]. There exists a discernible relationship between transformational leadership, characterized as a needs-supportive leadership style, and subordinates’ basic psychological needs and autonomous motivation [54]. This relationship could constitute one of the mechanisms through which transformational leadership impacts the quality and quantity of subordinates’ work [33]. 

In a recent study, Amor et al. investigated the relationship between transformational leadership behavior and subordinates’ work engagement based on the self-determination theory and psychological contract theory [55]. They also examined the partial mediating role of structural empowerment in this relationship. Their findings suggested that high-level transformational leadership could enhance subordinates’ sense of structural empowerment by providing them with access to information, opportunities, support, and ample resources, thereby promoting work engagement. 

This study, drawing from the theoretical framework of Amor et al., substitutes the psychological contract for structural empowerment as a mediating variable. This is based on the premise that the fulfillment of a psychological contract and psychological empowerment are interdependent and, as acknowledged by the authors, psychological empowerment is a precursor to structural empowerment [55]. However, the application of the psychological contract as a mediating variable in the research of transformational leadership is relatively rare. Chen et al. considered the psychological contract as a mediating variable in exploring the relationship between transformational leadership and subordinate emotional labor [28]. Therefore, by referencing Chen et al., this study incorporates the psychological contract as a mediating variable into the research framework to further elucidate the mechanisms of transformational leadership.

While Chen et al.’s design considering the psychological contract as a mediating variable offers theoretical support for this study, their research only accounted for relational and transactional psychological contracts, neglecting developmental psychological contracts, which are equally significant in this study. Developmental psychological contracts emphasize the mutual responsibilities of the employer and employee regarding future career success and development [56]. This includes job roles with developmental space, challenging job content, obtaining a sense of achievement and satisfaction in the job, and voluntarily taking on extra-role work tasks, such as the exchange and sharing of work skills [57]. 

These concepts align closely with the philosophy of group-focused transformational leadership, which strives to paint an appealing future vision, encourages subordinates’ participation in decision-making, collaboratively establishes group goals, and sets high performance expectations based on these goals [58]. A compelling vision can cause teachers to perceive their job roles as having developmental space [59]. Group goals prompt group members to voluntarily take on extra-role work tasks [60]. Setting high-performance expectations based on group goals makes teachers’ job content more challenging [61]. This challenge may stimulate teachers’ intrinsic motivation, encouraging them to utilize their abilities fully while accomplishing group goals [61]. Under the stimulation of intrinsic motivation, the degree of accomplishment of group goals may exceed initial expectations. This unexpected exceedance could provide teachers with a greater sense of achievement and satisfaction in their work [62].

Still, the study by Amor et al. treats transformational leadership as a single holistic construct, refraining from a segregated analysis of the respective impact mechanisms of group-focused and individual-focused leadership on essential psychological elements and job performance [55]. The majority of the prevailing literature attributes the fulfillment of psychological contracts at an individual level to individual-focused transformational leadership [29,63], particularly the notion that transformational leadership augments psychological contract satisfaction via certain individualized consideration behaviors [63]. 

These studies, to a certain degree, overlook the significant role that group-focused transformational leadership plays in bolstering internal group objectives [58] and fostering group cohesion [64]. Consequently, it is of importance to address this gap and illuminate the contribution of group-focused transformational leadership to the fulfillment of psychological contracts.

Lastly, Rousseau and Fried argue that cultural conditions exert a profound impact on theoretical constructs, necessitating a distinct approach when analyzing disparate countries [65]. With the rapid progression of reform and liberalization in the past few decades, China has undergone a pivotal transition from a planned economy to a market-oriented one, resulting in significant shifts in societal values [66]. Nevertheless, the imprint of a collectivist culture remains profound within the Chinese populace, influencing elements such as personality development, attitude formation, and behavioral norms to a considerable extent [67].

Hofstede extensively explored collectivism on the national cultural level, deeming it a vital dimension for cultural categorization and identifying China as a paradigmatic collectivist nation [68]. As research advanced, scholars shifted their focus towards the comprehension of collectivist culture at the individual and group levels. Among these, group collectivist orientation emerged as a group-level variable, representing an underlying set of norms and cultural ambiance within a group [69]. It accentuates the importance of harmonious relations within groups, encourages employees to internalize the group’s goals, promotes concern for the welfare of the group and colleagues, and underscores the subordination of individual goals to group objectives [70].

According to the concept of leader–follower value congruence [71], when subordinates’ values align with the leadership style in terms of ideology, subordinates are more likely to respond strongly to that leadership style, whereas the influence of this leadership style on subordinates is comparatively weaker when there is a divergence in values. Group-focused transformational leadership places importance on collective awareness and group objectives [72], encouraging group members to collaborate synergistically for the attainment of common goals [73]. Group members with a high degree of group collectivism tend to adhere to collective norms, prioritize group objectives, and actively participate in cooperative endeavors [74]. This alignment corresponds with the ideology of group-focused transformational leadership. Consequently, group members immersed in a high-level group collectivist atmosphere are more likely to internalize the leadership’s group values, leading to a manifestation of organizational citizenship behavior [75]. Conversely, group members in a low-level group collectivist atmosphere hold self-centered values [42], which can be viewed as incongruent with the ideology of group-focused transformational leadership. As a result, for such employees, the impact of group-focused transformational leadership on organizational citizenship behavior might be relatively weaker.

In the realm of education, teachers, shaped by various cultural value orientations, adhere to divergent behavioral norms. This divergence can precipitate substantial differences in the influence of certain factors on teachers’ organizational citizenship behavior [76]. To address this phenomenon and to respond to Bass’s proposition that transformational leadership initially conceptualized and measured within the individualistic context of the United States appears equally, if not more, applicable in the collectivist societies of Asia [44], this study incorporates collectivist culture as a moderating variable. This is achieved within the research framework to expand upon the model proposed by Amor et al. [55]. The incorporation is intended to shed further light on the boundary conditions pertinent to the impact of group-focused transformational leadership on teachers’ organizational citizenship behavior, thereby enhancing our comprehension of such behavior within Chinese educational institutions. This approach is significant, as, in a collectivist group, an emphasis is often placed on group cooperation and common interests. In such a context, subordinates may demonstrate a greater inclination to respect their leaders [40,77], aligning with the group values promoted by leadership [71], thus potentially amplifying the impact of transformational leadership.

To conclude, this study proposes a cross-level impact mechanism model of transformational leadership. The aim is to delve into two cross-level impact effects transitioning from the group level to the individual level: first, the influence of group-level, group-focused transformational leadership on teachers’ organizational citizenship behavior, and second, the mediating role that individual-level psychological contract fulfillment plays within this impact mechanism. The proposed theoretical model is depicted in Figure 1.

### 2.2. Group-Focused Transformational Leadership and the Organizational Citizenship Behavior of Teachers

Group-focused transformational leadership, operating under the paradigm of Average Leadership [73], primarily motivates members as a cohesive unit, imparting an influence that is homogeneously exerted on each individual within the group [78]. Conceptually, the behavior of group-focused transformational leaders affects all group members collectively rather than any single individual [79]. 

As identified by Podsakoff et al., there are four distinct facets of group-focused transformational leadership that could exert a comprehensive influence on subordinates [14]. Firstly, the dimension of articulating a vision accentuates the leader’s commitment to discern and instantiate new opportunities for the group. This facet seeks to inspire all followers by envisaging a compelling future vision, thereby influencing the group’s collective sense of mission. Furthermore, transformational leadership emphasizes the strong spiritual characteristics of leaders [80]. The second dimension stems from this, and it involves providing an appropriate model, setting an exemplar for all followers, and fostering alignment of subordinates with the values espoused by the leader [79]. The third dimension predominantly pertains to the establishment of group goals, fostering the acceptance of group goals among the members. Leaders, in this capacity, go to great lengths to create an ideal environment for cooperation among group members, persistently motivating them to strive towards the shared goals and interests of the group [14]. The final dimension is high performance expectations. Herein, leaders set forth high but realistic expectations for the group, in accordance with the prevailing circumstances [14].

Group-focused transformational leadership fosters elevated work motivation among group members by crafting and conveying a compelling group vision, conveying lofty accomplishments and values, sparking work enthusiasm, and imbuing work with greater intrinsic value [79]. Gradually, the collective group objectives are internalized into the personal goals of group members, leading to their identification with group goals and the realization that the achievement of group objectives is a prerequisite for realizing personal goals [10]. 

Consequently, this influence encourages group members to view work tasks through the lens of collective benefits and prioritize collective interests over personal ones. It motivates subordinates to transcend personal needs, contribute selflessly to collective interests, and ultimately exhibit enhanced organizational citizenship behavior [81].

Moreover, organizational citizenship behavior possesses certain social attributes [82]. The interpersonal ambiance and relational status within a group are often considered significant determinants of organizational citizenship behavior [83]. Group-focused transformational leadership accentuates communication and cooperation among members of teacher groups [64]. It focuses on the creation of a supportive, relaxed, and trustful environment within the teacher groups, fostering positive interpersonal interactions and relationships, and subsequently encouraging more organizational citizenship behavior among individual teachers [84]. Thus, based on the above analysis, this study proposes the following hypothesis:

**H1:** 
*Group-focused transformational leadership significantly and positively influences the organizational citizenship behavior of teachers.*


### 2.3. The Mediating Role of the Psychological Contract of Teachers

The concept of the psychological contract primarily pertains to the relationship between the organization and its employees, typically encompassing both the employee psychological contract and the organizational psychological contract [85]. Nonetheless, due to the difficulty in delineating the subject of the organizational psychological contract, the majority of contemporary research on psychological contracts is conducted primarily from the perspective of the employee psychological contract. This approach refers to employees’ subjective perceptions of mutual responsibilities and obligations between themselves and the organization [86]. In comparison to explicit contracts, psychological contracts, while implicit and informal, play a role analogous to explicit contracts, influencing mutual trust and understanding between employees and the organization [87]. 

The principal, who is the leader, serves as the main executor of organizational responsibilities and conveys organizational ideas and expectations to group members. In this process, the leadership style exhibited inherently affects employees’ perceptions of their individual and organizational responsibilities [88]. This may subsequently bolster their awareness of organizational citizenship.

Group-focused transformational leadership excels at portraying clear and challenging group goals to group members [19], using their own behavior to inspire group members [79]. This kind of leadership instills in group members the belief that the goals outlined by the leader can be achieved, and thereby turns challenging objectives into intriguing pursuits [89]. As a result, group members’ work enthusiasm might be boosted, and the group, as well as its members, may achieve more significant success than initially expected. 

Moreover, group-focused transformational leadership is committed to fostering collaboration among group members [31]. Group members incorporate group goals into their individual objectives, which could foster a strong sense of group identity [10]. Consequently, a congenial and unified group environment is established, group members could become more cohesive, and the group atmosphere becomes more harmonious [53]. 

The cultivation of a positive group atmosphere, coupled with the sense of accomplishment and satisfaction experienced by teachers in accomplishing challenging group goals, are stimulated by the leaders. According to self-determination theory, these external motivational factors can interact with employees’ intrinsic motivations [33,90], such as their desire for meaningful work and personal growth, and gradually become internalized as intrinsic motivations [91]. This is a dynamically constructed process that evolves through continuous interaction between teachers and school leaders, explaining how teachers gradually come to identify with extrinsic motivations and organizational expectations that align with their own intrinsic values and goals [49]. This process of internalization may not only promote high-quality communication between school leaders and teachers but also strengthen psychological contract fulfillment [85]. When teachers internalize the organization’s expectations and values, they are more likely to identify and adhere to the psychological contract, further enhancing high-quality exchange relationships between leaders and teachers [92].

The leader–member exchange theory, grounded in the principle of reciprocity, posits that when leaders act in the interests of both parties in the labor–management relationship and subordinates derive benefits aligned with their psychological expectations from this process, the subordinates, in reciprocation, will exhibit positive behaviors and attitudes towards both the leader and the group [93].

From this standpoint, the relationship between superiors and subordinates fundamentally constitutes a contractual relationship [32]. Transformational leadership, acknowledging the capabilities of employees, proposes ambitious and challenging objectives for the group, expressing high performance expectations for all group members, and strives to engender a harmonious and collaborative group environment. By doing so, transformational leadership elevates the level of expectations and confidence regarding task completion for both the group and individual group members [52]. This phenomenon can be perceived as fulfilling the organization’s interpersonal and developmental responsibilities.

In providing a compelling vision for group members and offering developmental opportunities, transformational leadership accomplishes the organization’s transactional responsibilities [30]. When employees perceive these fulfilled responsibilities and these surpass their intrinsic expectations, leaders gain the employees’ trust, fostering the transition of leader–employee relationships from transactional exchange to social exchange [93].

According to the reciprocity principle of the social exchange theory, employees, feeling obligated to reciprocate the benefits they have received, engage in proactive behaviors such as organizational citizenship behavior [32]. Following the aforementioned discussion, this research proposes the ensuing hypothesis:

**H2:** 
*The psychological contract of teachers plays a mediating role between group-focused transformational leadership and the organizational citizenship behavior of teachers.*


### 2.4. The Moderating Role of Collectivism

Apart from comprehending leadership efficacy by examining the impact mechanism of group-focused transformational leadership on organizational citizenship behavior, this study also adopts a contingent perspective to analyze group-focused transformational leadership. It posits that leadership effectiveness is contingent not only on the psychological or physiological attributes of the leader or subordinates, but also on situational factors. Traditional Chinese culture, with its core focus on collectivism, is a dominant element in the context of the Chinese cultural background. This culture accentuates the equitable distribution of resources and benefits, suggesting that Chinese employees are more inclined towards leaders who prioritize the welfare of the entire group rather than that of individual members [94].

Collectivism is a cultural attribute that accentuates the accomplishment of collective interests over individual ones [95]. In a collectivist milieu, individuals typically prioritize collective interests, uphold group cohesion, make personal sacrifices for the group, foster the development of other group members, and are more inclined to conform to authority [96] as a means to facilitate their own socialization process [97]. Complementary to this, social identity theory posits that employees generally exhibit social identification to bolster self-esteem and to seek their social position or corresponding benefits [98]. Thus, it can be conjectured that variations in the collectivist atmosphere might influence the cognition of group members, thereby impacting their behavior.

Schaubroeck et al. conceptualized collectivism at the group level and introduced the notion of group collectivist orientation [45]. This refers to the extent to which a group prioritizes the preservation of social and interpersonal relationships over personal interests, emphasizing norms of mutual assistance, affection, and support. In their study, Walumbwa and Lawler discovered a moderating role of collectivism in the relationship between transformational leadership and work-related outcomes [99]. This study postulates that within a high-level collective group, teachers tend to appreciate group goals, comply with authority, and value group cohesion. Consequently, the effectiveness of transformational leadership is amplified. Based on the preceding analysis, the following hypothesis is proposed:

**H3:** 
*Collectivism significantly and positively moderates the cross-level relationship between group-focused transformational leadership and the individual-level organizational citizenship behavior of teachers.*


## 3. Methods

### 3.1. Research Context

Teachers in Chinese secondary schools hold multifaceted roles that transcend the conventional confines of imparting subject-specific knowledge and facilitating students’ understanding of pertinent content and knowledge schemas. They are integral in shaping moral and ideological education, guiding students towards the formation of appropriate values, and fostering comprehensive development [100]. Furthermore, Chinese society emphasizes the importance of education and regards it as a crucial pathway to elevate one’s social status [101]. Consequently, the teaching profession in China is imbued with unique significance and value within its cultural context [102]. This cultural tradition transforms teachers into not only knowledge disseminators, but also societal leaders and moral exemplars [101]. Teachers often uphold high moral standards and, likewise, bear significant societal expectations and responsibilities. These elevated expectations and moral constraints can, to some extent, influence teachers’ work attitudes and behaviors [103].

Regarding compensation design, most schools follow an egalitarian approach in basic salary distribution, resulting in relatively minor differences in basic salary amounts among teachers [104]. The allocation of performance-based salary is relatively straightforward and primarily linked to students’ academic achievements. This includes quantifiable metrics such as pass rates, excellence rates, and rates of students advancing to higher education institutions [105].

Schools typically establish various teaching and research groups based on subject categories. Group members share a common workspace and hold seminars several times a week, providing ample opportunities for daily communication among members. As a result, interpersonal relationships among members of these teaching and research groups tend to be positive and strong.

### 3.2. Sample and Procedure

This study aimed to investigate the relationship between transformational leadership and organizational citizenship behavior among secondary school teachers. Initially, it is essential to note that subjects such as arts and sports are not included in the Chinese education examination system for further studies. Consequently, teachers instructing these subjects often face less pressure than those teaching core subjects (Chinese, Mathematics, English, Geography, Politics, History, Physics, Chemistry, and Biology) [106], which might lead to a lower incidence of organizational citizenship behavior [107]. Therefore, the sample selection for this study was restricted to teachers engaged in instructing subjects that fall within the purview of the Chinese education examination system.

Additionally, variations in organizational citizenship behavior could be influenced by the type of school. Teachers in ordinary secondary schools might be inclined to exhibit organizational citizenship behavior primarily through their students’ academic achievements, while teachers in secondary vocational schools might focus more on students’ vocational skills or employability. Hence, this study confined its sample selection to teachers from ordinary secondary schools.

Moreover, holding administrative or academic positions has been evidenced to influence teachers’ perception of transformational leadership or organizational citizenship behavior [108]. In view of this, the samples chosen for this study were restricted to full-time secondary school teachers who do not hold any administrative or academic positions.

Due to the cross-level design of this study, both group-focused transformational leadership and collectivism were considered group-level variables, which were expected to be aggregated from individual-level. To assess whether the data obtained from the scales were suitable for cross-level analysis, a small-scale pilot investigation was conducted before the formal investigation. The pilot investigation was conducted in 10 public high schools in Guizhou Province, China. With the consent of the school principals, the investigators distributed the paper questionnaires to participating teachers in their workplaces and provided them with face-to-face explanations of the purpose of the research, questionnaire instructions, and assurances of anonymity. All participants completed the questionnaires on-site. After completion, the questionnaires were collected by the investigators to ensure the effectiveness and confidentiality of the investigation. The pilot investigation covered 10 schools and 40 teaching and research groups, with a total of 407 questionnaires distributed and 392 collected. Among them, there were 383 valid questionnaires, resulting in an effective sample return rate of 94.1%. The pilot investigation data were used to perform aggregation tests on the group-level constructs involved in the study.

Three indicators, namely, the intraclass correlation coefficients (ICC (1) and ICC (2)) and the reliability within groups (r_wg_), were selected for the aggregation test in the pilot investigation. Both group-level variables met the aggregation criteria, with r_wg_ values above 0.70, ICC (1) values above 0.05, and ICC (2) values above 0.50. The results of the pilot investigation are presented in Table 1.

Upon defining the research population, systematic sampling, cluster sampling, and simple random sampling techniques were employed to distribute paper questionnaires across 31 provinces in China. The specific steps are outlined as follows: First, the investigators utilized the ratio of the number of middle school teachers in each province to the total number of middle school teachers in China as a reference standard [109]. The number of schools selected for this study was computed to be 100 times this ratio. For instance, in Jiangsu Province, which hosts 364,809 middle school teachers, accounting for 5.50% of the total number of middle school teachers nationwide, six schools were selected for sampling.Subsequently, we randomly selected schools from the list of middle schools provided by the Chinese Ministry of Education in accordance with the aforementioned proportions. Once the list of chosen schools was established, the personnel departments of these schools were contacted to introduce the purpose, anonymity, significance, and practical implications of this study for school development. Upon obtaining consent from the school personnel departments to participate in the research, questionnaires and informed consent forms were mailed to the respective schools. Due to the nested structure of the data involved in this study and an anticipated investigation of 100 teaching and research groups, both data aggregation and random effects analysis needed to meet a minimum ratio of at least 100 /10 (a minimum of 10 samples from within each of the 100 groups are required) [110]. Therefore, the personnel in each school were asked to randomly distribute the questionnaire to any 15 teachers in any teaching and research group that met the sample criteria.Finally, after the teachers completed the questionnaire, the personnel departments of each school were responsible for collecting the questionnaires and mailing them back to the investigators, with postage paid by the investigators.

The distribution of the questionnaire was carried out from January to March 2023, during which a total of 1560 questionnaires were disseminated. Upon exclusion of invalid questionnaires, a total of 1162 valid questionnaires were ultimately collated from 104 schools, yielding an overall recovery rate of 74.5%. Every group in the sample contributed at least 10 questionnaires, thus adhering to the aggregation standards stipulated in cross-level research [110]. The detailed demographic characteristics of the samples are shown in Table 2.

### 3.3. Measurement Tools

This study utilized all variables measured through a 7-point Likert scale, varying from 1 (strong disagreement) to 7 (strong agreement). For the ease of respondent comprehension, all questionnaires were delivered in Chinese. Each English scale adhered to the standard protocol of translation and back-translation to minimize potential errors arising from translation [111]. Complete scale is presented in the Appendix B.

#### 3.3.1. Group-Focused Transformational Leadership

To measure group-focused transformational leadership, this study incorporated the scale adapted by Zhang et al., predicated upon the hierarchy of transformational leadership as postulated by Podsakoff et al. [14,79]. This scale encompasses four dimensions—articulating a vision, providing an appropriate model, fostering the acceptance of group goals, and high-performance expectations—consisting of 15 items. In the research of Zhang et al., the Cronbach’s α coefficient for group-focused transformational leadership was reported as 0.95 [79]. In addition, the scale was also validated in a Chinese cultural context [112]. Therefore, the scale was selected to measure group-focused transformational leadership in this study, and the items were modified to fit the educational context.

#### 3.3.2. Psychological Contract of Teachers

To measure the psychological contract fulfillment of teachers, this study utilized the questionnaire formulated by Han et al. [57], divided into two subscales: school’s obligations and teacher’s obligations. Given the study’s premise that secondary school teachers enhance their psychological contracts with the school under the influence of the principal’s transformational leadership, only the subscale of teacher’s obligations to the school was employed in the measurement of the psychological contract fulfillment of teachers. This subscale includes three dimensions—transactional obligations, developmental obligations, and relational obligations—with a total of 10 items. The Cronbach’s α coefficient of this scale was 0.86, demonstrating commendable reliability.

#### 3.3.3. Organizational Citizenship Behavior of Teachers

This study aimed to explicate the mechanisms driving the development of organizational citizenship behavior among secondary school teachers. The scale formulated by Chou et al. was adopted, which stratifies organizational citizenship behavior into five distinct dimensions (student emotional counseling, student employment support, teacher teaching support, self-enhancement and development, and participation in social welfare) [75]. This scale contains a total of 16 items and demonstrated robust reliability, as indicated by a Cronbach’s α coefficient of 0.91. In consideration of the specific target population of this research, secondary school teachers, a particular adjustment to the questionnaire was seen as necessary. The teaching roles of these professionals demonstrate a weak correlation with student employment support, a category within the scale. Consequently, two items pertaining to student employment support were removed, preserving the remaining 14 items for inclusion in this study.

#### 3.3.4. Collectivism

This study measured collectivism using the scale devised by Schaubroeck et al. [45]. All items were reverse-scored, amounting to 3 items. The Cronbach’s α coefficient was 0.89, and this scale was recently introduced and applied to the Chinese context by Peng et al., validating it as reliable [113].

#### 3.3.5. Control Variables

In accordance with a similar study [114], demographic variables that could potentially influence the hypothesized relationships were controlled. These variables encompassed gender, age, teaching experience, education background, and professional title. The method of numerical encoding was employed to code the control variables.

Incorporating these variables into the analysis assisted in ensuring that the observed relationships were not confounded by these factors, thereby fortifying the validity and reliability of the survey outcomes.

### 3.4. Data Analysis

#### 3.4.1. Reliability and Validity

Using SPSS 25.0 and Mplus 8.3 software, Cronbach’s α (CA) values and composite reliability (CR) values were employed to assess the reliability of the scales used in this study. The test outcomes are displayed in Table 3. The CA values all exceeded the reference standard of 0.70. Moreover, the CR values for each scale also surpassed the reference standard of 0.70 (as shown in Table 3). Based on these evaluations, the measurement scales demonstrated substantial reliability [115].

Moreover, the validity was assessed in three ways. Firstly, confirmatory factor analysis (CFA) was conducted on each variable using Mplus 8.3 software. The convergent validity of the scales was evaluated based on two indicators: factor loadings and Average Variance Extracted (AVE). Factor loading values greater than 0.5 for each measurement item and AVE values exceeding 0.5 both denoted substantial convergent validity of the measurement model [116]. All factor loading values were greater than 0.5, ranging from 0.707 to 0.901. As demonstrated in Table 3, AVE values also fulfilled the requirements, proving good convergent validity of the scales. Secondly, discriminant validity of the scales was determined by comparing AVE values with correlation coefficients. Good discriminant validity of the scales was evident when the square root of each latent variable’s AVE value was larger than its correlation coefficient with other latent variables [117]. As depicted in Table 3, the square root of AVE values for each latent variable was larger than its correlation coefficient with other latent variables, signifying high discriminant validity of the scales. Lastly, following the suggestion of Li et al. [72], the discriminant validity of group-level variables and individual-level variables was tested separately. The results are illustrated in Table 4. According to Wu [118], the criteria for evaluating the goodness of fit of the measurement model were: χ^2^/df less than 5, RMSEA less than 0.08, and CFI and TLI both larger than 0.9. According to the test results, the fit of the two-factor model at both the group level and individual level surpasses that of the single-factor model, indicating good distinctiveness of the variables defined in this study at both the group and individual levels.

#### 3.4.2. Common Method Bias

As the data in this study originated from the same participants, the common method variance (CMV) could potentially undermine the study’s validity. To test for CMV, Harman’s single factor test was performed. The factor analysis of the study resulted in six eigenvalues greater than 1, cumulatively accounting for 74.38% of the explained variance. The explanatory power of the first factor was only 25.29%, suggesting that CMV did not critically affect this study [119].

#### 3.4.3. Aggregation and Analytical Strategies

In addition, considering the nested data structure, this study employed a multilevel structural equation model, encompassing group-level variables (group-focused transformational leadership and collectivism) and individual-level variables (psychological contract and organizational citizenship behavior). Therefore, we demonstrated the appropriateness of data aggregation by measuring r_wg_, ICC (1), and ICC (2).

Given the involvement of individual-level mediating variables and dependent variables in the research model, as well as independent and moderating variables at the group level, a multilevel analysis approach was required to test the regression relationships among multiple model variables. Thus, Mplus 8.0 software was utilized to construct a multilevel structural equation model to examine the hypotheses. Furthermore, in terms of testing the mediating effects, this study followed the logic of traditional methods to assess the mediating effects in this research. Regarding the examination of moderating effects, this study tested the moderating effect of collectivism by constructing the interaction term between group-focused transformational leadership and collectivism. The equations for testing the multilevel hypotheses in this study are as follows (See Table 5).

## 4. Findings

### 4.1. Aggregation Test

This research considers group-focused transformational leadership and collectivism as group-level variables. In terms of aggregation testing, it necessitates the examination of intra-group homogeneity and inter-group heterogeneity for these variables. Furthermore, the dependent variable, organizational citizenship behavior of teachers, although measured at the individual level, also embodies differences at the group level. Hence, it is essential to examine its intra-group homogeneity and inter-group heterogeneity. The test results are exhibited in Table 6. The results reveal that the mean r_wg_ values for group-focused transformational leadership and collectivism are both above the standard of 0.70, meeting the basic standard of within-group rating consistency. Furthermore, the values for both ICC (1) and ICC (2) exceed the recommended critical values of 0.05 and 0.5, thereby fulfilling the requirements for data aggregation [120]. Consequently, it is appropriate to aggregate group-focused transformational leadership and collectivism to the group level.

### 4.2. Hypothesis Testing

#### 4.2.1. Test of the Main Effect

Initially, a null model was constructed for teacher organizational citizenship behavior and the psychological contract of teachers, yielding ICC (1) values of 0.696 and 0.317, respectively. This suggests that the variance in different organizational citizenship behavior of teachers and psychological contract of teachers is explained by between-group differences, accounting for 69.6% and 31.7% of the variance, respectively, and displays substantial between-group variance. Thus, organizational citizenship behavior of teachers and the psychological contract of teachers can be used as dependent variables for cross-level analysis [120]. On this basis, using teacher organizational citizenship behavior as the dependent variable and individual-level control variables (gender, age, teaching experience, educational level, and professional title), Model 1 was constructed to test the impact of control variables on the organizational citizenship behavior of teachers. Building upon Model 1, group-focused transformational leadership was integrated into the regression equation, forming Model 2, which tested the main effect of group-focused transformational leadership on organizational citizenship behavior of teachers. The regression analysis revealed a regression coefficient of 0.881 (*p* < 0.001) for group-focused transformational leadership’s impact on the organizational citizenship behavior of teachers, signifying a substantial cross-level positive influence. Thus, Hypothesis H1 was validated.

#### 4.2.2. Test of the Cross-Level Mediating Effect of the Psychological Contract of Teachers

Upon the basis of Model 1, the individual-level psychological contract was incorporated to establish Model 3, with the aim of exploring the impact of the psychological contract on the organizational citizenship behavior of teachers. Regression analysis revealed a coefficient of 0.697 (*p* < 0.001) for the impact of the psychological contract on the organizational citizenship behavior of teachers, signifying a significant positive influence of the psychological contract on the organizational citizenship behavior of teachers.

As group-focused transformational leadership is a group-level variable, a linear regression analysis was conducted where the individual-level psychological contract was posited as the dependent variable and group-focused transformational leadership was used as the independent variable. Consequently, Model 4 was constructed to evaluate the cross-level impact of group-focused transformational leadership on the psychological contract. The analysis revealed a coefficient of 0.876 (*p* < 0.001) for group-focused transformational leadership on the psychological contract, thus denoting a significant positive effect of group-focused transformational leadership on the psychological contract.

Adopting the procedure for cross-level mediation proposed by Zhou and Pan [121], the psychological contract was introduced as a mediator into the regression equation based on Model 3, resulting in the construction of Model 5. This model was designed to scrutinize the cross-level mediating role of the psychological contract between group-focused transformational leadership and the organizational citizenship behavior of teachers. The findings demonstrated a coefficient of 0.414 (*p* < 0.001) for the impact of group-focused transformational leadership on the organizational citizenship behavior of teachers. This coefficient, when compared to the corresponding value of 0.881 (*p* < 0.001) in Model 2, implies a considerable reduction in the coefficient of the effect of the independent variable on the dependent variable due to the inclusion of the mediator. This infers that the psychological contract imparts a partial mediating influence between group-focused transformational leadership and the organizational citizenship behavior of teachers. Consequently, Hypothesis H2 is supported. Detailed cross-level regression analysis results are presented in Table 7.

#### 4.2.3. Test of the Moderating Effect

Collectivism was considered as a moderating variable, acting on the regression effect of group-focused transformational leadership on the organizational citizenship behavior of teachers, to analyze whether collectivism plays a moderating role. In accordance with the recommendations of Memon et al. [122], the interaction term was constructed by multiplying in sequence according to the load value relationship between the moderating variable and the group-level independent and mediating variables. The standard for establishing a moderating effect was whether the impact of the interaction term on the organizational citizenship behavior of teachers was significant.

The analysis revealed a significant positive regression impact of the interaction term between group-focused transformational leadership and collectivism on the organizational citizenship behavior of teachers, with a standardized regression coefficient β = 0.357 and a significance test result of *p* < 0.05. This suggests that collectivism plays a significant positive moderating role in the regression impact of group-focused transformational leadership on the organizational citizenship behavior of teachers.

Furthermore, to better reflect the role of the moderating effect, the moderating variable was divided into two groups—high and low—by adding or subtracting a standard deviation from the mean [123]. The Johnson–Neyman method was used to calculate and draw the simple slope graph of the moderating effect through a simple slope analysis [124], as shown in Figure 2.

When the moderating variable of collectivism is low (M − SD), the regression coefficient of group-focused transformational leadership on the organizational citizenship behavior of teachers is 0.284, with a significance test result of *p* = 0.089 > 0.05. When the moderating variable of collectivism is high (M + SD), the regression coefficient is 0.858, with a significance test result of *p* < 0.001. This implies that as the value of the moderating variable increases, the regression impact of group-focused transformational leadership on the organizational citizenship behavior of teachers continually strengthens, changing from insignificant to significant. Furthermore, the difference in regression coefficients under high and low moderating variable values is significant (b = 0.574, *p* = 0.012 < 0.05), and hence, Hypothesis H3 is confirmed.

## 5. Discussion and Conclusions

The primary aim of this study was to address several critical questions: does school transformational leadership, in the form of the principal’s group-focused transformational leadership behavior, enhance the level of teacher’s organizational citizenship behavior? Can it bolster individual-level teacher psychological contract fulfillment, thereby facilitating an elevated level of organizational citizenship behavior of teachers? Is there a spillover effect of this group-level influence from the group to the individual level? Additionally, does collectivism at the group level strengthen the relationship between group-focused transformational leadership and individual-level organizational citizenship behavior?

In response to these questions, a cross-level analysis model was constructed through theoretical deduction and empirical testing, with the intent of examining the specific mechanisms through which group-focused transformational leadership impacts the organizational citizenship behavior of teachers.

The findings indicate that: (1) the principal’s group-focused transformational leadership exerts a cross-level positive influence on the individual-level organizational citizenship behavior of teachers; (2) individual-level psychological contract fulfillment plays a pivotal cross-level mediating role between group-focused transformational leadership and the organizational citizenship behavior of teachers; and (3) collectivism at the group level positively moderates the process in which group-focused transformational leadership influences the individual-level organizational citizenship behavior of teachers. When a group is characterized by a high level of collectivism, the impact of group-focused transformational leadership on the organizational citizenship behavior of teachers is more profound.

Organizational citizenship behavior is voluntary and unlikely to occur without strong intrinsic motivation [8]. However, when group members strive for common goals under the stimulus of group-focused transformational leadership, they tend to make greater contributions to the group and therefore are more likely to engage in proactive behaviors in anticipation of improved group performance.

Secondly, utilizing the framework of self-determination theory, this study verifies the cross-level mediation role of individual-level psychological contract fulfillment in the relationship between group-focused transformational leadership and the organizational citizenship behavior of teachers. Psychological contract theory posits that the parties in an employment relationship may have different perceptions of each other’s rights and obligations [86] and that an employer’s obligations influence employee obligations and behavior, taking precedence in contract performance [50]. 

Regardless of the type of psychological contract, employees perceive it as a form of a socio-emotional resource, and they tend to reciprocate the efforts of supportive employers with proactive behavior [93]. Group-focused transformational leadership is committed to fostering a sense of unity among all group members, leading to a closely knit group. Such behavior can be perceived by employees as leaders fulfilling their emotional obligations to the group in order for the group to achieve its objectives. In return, employees may engage in proactive behaviors that are beneficial to collective interests, such as organizational citizenship behavior [32].

Furthermore, regarding the relationship between group-focused transformational leadership and the organizational citizenship behavior of teachers, this study confirms the positive moderating role of collectivism at the group level, providing empirical support for the theoretical hypothesis proposed by Nasra and Heilbrunn [40]. In groups with a high level of collectivism, teachers are more inclined to respect leadership and are greatly influenced by the leadership style [77]. Moreover, their personal goals highly align with the group goals, viewing their individual goals as subsets of the group goals [96]. Therefore, under the moderating effect of collectivism, the influence of group-focused transformational leadership on the organizational citizenship behavior of teachers is enhanced.

### 5.1. Theoretical Implications

Based on the context of Chinese school organizations, the first contribution of this study is the proposition of a cross-level model of group-focused transformational leadership and organizational citizenship behavior, supported by empirical evidence. This study responds to calls for a fine-grained investigation of transformational leadership [19,21,25,26]. This study contributes to transformational leadership research by explaining its impact mechanism from a cross-level perspective [18]. Most previous research primarily focused on clarifying the inherent pattern of a single construct transformational leadership style with organizational citizenship behavior, with scant attention paid to the effect of group-focused transformational leadership on organizational citizenship behavior. This study finds that group-focused transformational leadership has a cross-level impact on the individual-level organizational citizenship behavior of teachers, providing a detailed explanation of how transformational leadership motivates subordinates. 

Amor et al. proposed a theoretical model based on the self-determination theory framework, suggesting that transformational leadership influences subordinate work engagement through the mediating role of structural empowerment [55], and Chen et al. explored the relationship between transformational leadership and subordinate emotional labor with the psychological contract as a mediating variable. Building on these studies [28], this study, from the perspective of differentiated transformational leadership, further explores the impact mechanism of group-focused transformational leadership, enriching theoretical research on the relationship between leadership style and organizational citizenship behavior. The second contribution of this study is that it examines the mediating role of psychological contract fulfillment. This study examines the cross-level effects of group-focused transformational leadership on the psychological contract and organizational citizenship behavior, which to some extent explains the dynamic interaction between groups and individuals [125]. Regarding how these effects occur, this study goes beyond previous studies that only focused on the mediating effects of leadership and outcomes at a single level (either individual or group level) [126,127], demonstrating the important cross-level and “spillover” effects of group-focused transformational leadership on individual-level psychological contracts and organizational citizenship behavior.

In accordance with self-determination theory, there is a process of internalization bridging the gap between extrinsic motivation and intrinsic motivation [50]. This suggests that external factors can gradually evolve into internal factors, thereby influencing individuals’ deeper intrinsic motivations [90]. Transformational leadership plays a pivotal role in fostering organizational citizenship behavior among subordinates by igniting their intrinsic motivation [33]. Here, extrinsic motivation can be understood as the external incentives provided by leaders, while internal beliefs encompass subordinates’ intrinsic motivation, including their passion for their work, their need for autonomy, and the satisfaction of their basic psychological needs [34].

Transformational leaders, through their inspirational behaviors and leadership styles, have the power to encourage employees to internalize these incentives, thereby enhancing their intrinsic motivation [34]. This, in turn, leads to greater displays of organizational citizenship behavior [41]. In this process, psychological contract fulfillment could serve as a potential intermediary [86]. Psychological contract fulfillment reflects whether employees are content with the organization’s adherence to their psychological contract [85], and it can be seen as an expression of internal motivation since contented employees are more driven to support the organization’s objectives [63]. When employees perceive their leaders’ transformational leadership style and sense the motivation and support provided by these leaders, they are more likely to experience psychological contract fulfillment [31], consequently fostering a greater willingness to exhibit organizational citizenship behavior [32].

Finally, the third contribution of this study is that it tests the moderating effect of collectivism at the group level, expanding the boundary conditions of leadership research and responding to Wang and Howell’s call for research on the moderating mechanisms behind dual-level transformational leadership [10]. Currently in leadership research, studies have proven the moderating role of employees’ collectivist tendencies in the relationship between transformational leadership and organizational citizenship behavior [41,42]. These studies focus on using individual-level values as boundary conditions for leadership’s influence on subordinate organizational citizenship behavior, ignoring the influence of the situation on their behavior and attitudes. This study, from the perspective of group member interaction, treats collectivism at the group level as a trait activation factor, further explaining the relationship between group-focused transformational leadership and organizational citizenship behavior.

However, in contrast to the findings presented in this study regarding the effectiveness of collectivism in enhancing group-focused transformational leadership, Ma et al. argue differently in their research [128]. They suggest that within highly collectivistic new ventures, members may prioritize a sense of relatedness. When these relatedness needs are fulfilled, the influence of group collectivism on the effectiveness of transformational leadership may weaken. This conflicting conclusion could be attributed to differences in organizational contexts. In cultures with a strong emphasis on collectivism, decision-making typically involves a collective process rather than being solely determined by leaders [77]. Leaders may be required to invest more time in building consensus and coordinating group members to reach a decision. In new ventures operating within rapidly changing markets, this collective decision-making process might potentially hinder the speed and adaptability of leadership in driving transformative changes. Conversely, in educational organizations, leaders who actively listen to teachers’ opinions and respond patiently may find it easier to secure teachers’ support and trust [24]. This, in turn, can stimulate greater teacher engagement in pursuing shared objectives, ultimately augmenting the effectiveness of leadership [42].

### 5.2. Practical Implications

From the perspective of self-improvement, school leaders should focus on cultivating their ability to motivate the entire group. At the group level, school leaders could promote teachers’ organizational citizenship behavior by providing support in four key areas:

Firstly, school leaders should articulate a clear and compelling vision to group members and imbue it with meaning. This fosters member identification with the group, helps them internalize group goals as personal objectives, and stimulates a sense of shared ideals and values, thereby inspiring selfless dedication among members. Maxwell posits that emotions precede rationality, emphasizing the importance of a group vision in promoting organizational citizenship behavior among group members [129].

Furthermore, school leaders should set high performance expectations for group members to inspire passion and ambition in relation to their work, unlock their potential, and encourage continuous improvement in their work. In the implementation process, school leaders should avoid the “one-step-to-high-performance expectations” approach, as it can create immense work pressure for group members and even undermine their de-termination and confidence. School leaders can break down high performance expectations into several challenging sub-goals so that group members can gradually achieve them in a sequential and progressive manner.

Additionally, school leaders should lead by example, uphold high personal standards, enhance their professional skills, and strengthen their personal professional integrity. They should serve as role models for group members who strive for continuous learning. School leaders should also dedicate themselves to supporting teachers in pursuing group goals and creating the necessary conditions for teamwork within the teacher group [64]. This helps teachers perceive the contributions they make in helping to accomplish group goals, thereby achieving psychological contract satisfaction.

Finally, this study found an important positive moderating effect of a collectivistic atmosphere on the relationship between group-focused transformational leadership and the organizational citizenship behavior of teachers. This also holds significant implications for school management practices. In practical management, school leaders can foster the development of organizational citizenship behavior by creating a collectivistic atmosphere. Particularly in group environments with close collaboration, a collectivistic atmosphere can more effectively mobilize teachers’ organizational citizenship behavior, thereby enhancing the overall group performance and efficiency. Consequently, in practical management, school leaders can take various measures to cultivate a collectivistic atmosphere, such as establishing common group goals, strengthening group communication, and enhancing group cohesion, among other strategies.

## 6. Limitations and Future Research

Given various constraints and time limitations, this study has certain limitations that necessitate further exploration and improvement in future research. 

Firstly, the cross-sectional design of this study does not provide robust and clear causal inferences for the research findings, necessitating a cautious interpretation of the results. Future research should employ a longitudinal design to study the causal relationship between group-focused or differentiated transformational leadership and organizational citizenship behavior more accurately.

Secondly, while the confirmatory factor analysis in this study suggests that all constructs in this study are relatively independent, all variables were gathered through teachers’ self-reports, which may have introduced common method variance [130]. To mitigate this variance, future research could gather data through multiple time stages to provide more rigorous evidence.

Thirdly, this study explored the impact mechanism of group-focused transformational leadership on organizational citizenship behavior in detail. Leadership, however, may exist at multiple levels, including the group level and individual level [18]. In future research, the theoretical framework of Wang and Howell can be employed to construct a dual-level transformational leadership influence mechanism model [10]. This would explore the impact of transformational leadership on organizational citizenship behavior from a more comprehensive perspective.

Lastly, this study was conducted in a Chinese cultural context, which both serves as an advantage and a limitation. It provides unique insights for theoretical construction but might also limit the universality of the research findings. Despite previous research demonstrating consistency across different cultures [42], future research in other cultures is encouraged. This would enrich the application of transformational leadership under various boundary conditions and ensure its broader applicability and relevance.

## Figures and Tables

**Figure 1 behavsci-13-00848-f001:**
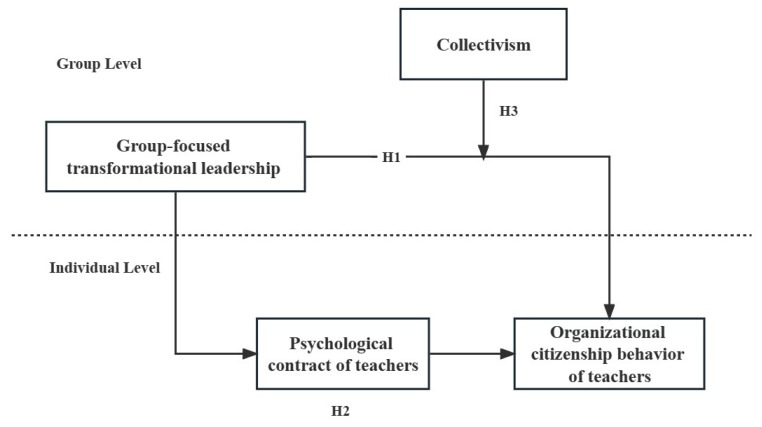
Theoretical model of this study. Notes: The upper dotted line denotes the group level, whereas the lower dotted line denotes the individual level.

**Figure 2 behavsci-13-00848-f002:**
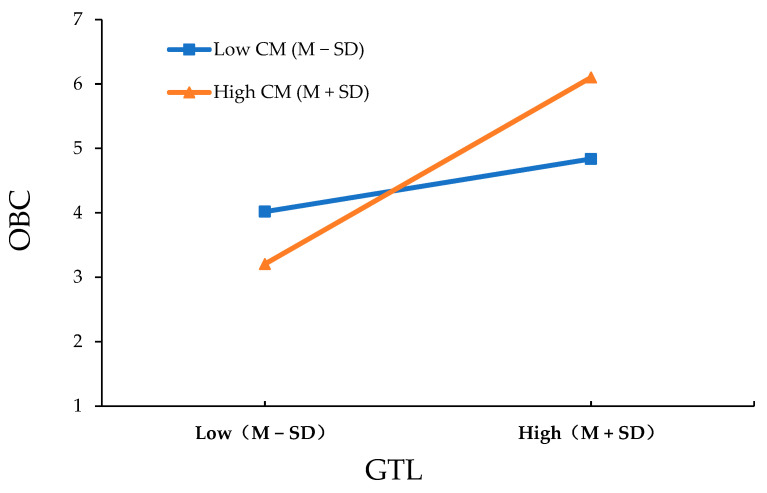
Collectivism as a moderator of the relationship between group-focused transformational leadership and the organizational citizenship behavior of teachers. Notes: GTL = group-focused transformational leadership; OCB = organizational citizenship behavior of teachers; CM = collectivism.

**Table 1 behavsci-13-00848-t001:** Results of aggregation test for pilot investigation.

Variables	r_wg_	ICC (1)	ICC (2)
GTL	0.952	0.446	0.883
CM	0.763	0.556	0.710

Notes: GTL = group-focused transformational leadership; CM = collectivism.

**Table 2 behavsci-13-00848-t002:** The demographic characteristics of the samples.

Variable	Category	Number	Percentage (%)
Gender	Male	612	52.7
	Female	550	47.3
Age	Under 28	149	12.8
	28–37	564	48.5
	38–47	348	29.9
	48 and above	101	8.7
Teaching Experience	Less than 3 years	108	9.3
	3–9 years	371	31.9
	9–15 years	393	33.8
	15 years and above	290	25
Education	College and below	146	12.6
	Bachelor’s degree	882	75.9
	Master’s degree	103	8.9
	Doctor’s degree	31	2.7
Professional Title	Junior and unrated	320	33.5
	Intermediate	643	48.2
	Associate Senior and Senior	199	18.3
School Level	Junior high school	362	31.1
	High school	800	68.9
Sample Size	Group level	104	
	Individual level	1162	

**Table 3 behavsci-13-00848-t003:** Descriptive statistics, correlations, and reliabilities.

Variables	Mean	SD	1	2	3					Cronbach α	CR	AVE
Group level (n = 104)												
1. Group size	11.173	1.038										
2. GTL	4.576	1.182		** *0.738* **						0.947	0.943	0.542
3. CM	4.796	1.443		−0.079 **	** *0.859* **					0.913	0.914	0.779
Individual Level (n = 1162)	Mean	SD	1	2	3	4	5	6	7			
1. Gender	1.47	0.500										
2. Age	2.35	0.810	−0.244 **									
3. Teaching Experience	2.74	0.936	−0.200 **	0.526 **								
4. Educational Background	1.96	0.396	0.142 **	−0.154 **	−0.101 **							
5. Professional title	1.85	0.704	−0.217 **	0.513 **	0.467 **	−0.103 **						
6. PCT	5.166	1.457	0.036	0.000	0.017	0.011	0.032	** *0.825* **		0.953	0.955	0.68
7. OCB	4.609	1.465	0.039	−0.007	−0.003	−0.021	−0.017	0.676 **	** *0.861* **	0.975	0.975	0.738

Notes: GTL = group-focused transformational leadership; PCT = psychological contract of teachers; OCB = organizational citizenship behavior of teachers; CM = collectivism; ** indicates significant correlation at the 0.01 level. Values that are emboldened and in italics are the square root of AVE values. Other values are the correlation coefficients of each variable.

**Table 4 behavsci-13-00848-t004:** Results of discriminant validity test.

Model	χ^2^	df	χ^2^/df	RMSEA	CFI	TLI
Group level						
Two-factor model	38.048	13	2.927	0.041	0.995	0.992
Single-factor model (GTL + CM)	2440.435	14	174.317	0.386	0.497	0.245
Individual level						
Two-factor model	22.293	13	1.715	0.025	0.999	0.999
Single-factor model (PTC + OCB)	2176.363	14	155.455	0.365	0.796	0.694

Notes: GTL = group-focused transformational leadership; PCT = psychological contract of teachers; CM = collectivism; OCB = organizational citizenship behavior of teachers.

**Table 5 behavsci-13-00848-t005:** Equations for testing the multilevel hypotheses in this study.

Null Model	
$CB={\beta_{0}+\beta }_{1}x_{1}+\beta_{2}x_{2}+\beta_{3}x_{3}+\beta_{4}x_{4}+\varepsilon_{1}$	Model 1
Level **1**	
$OCB=\beta_{1}+\beta_{11}\times PCT+\varepsilon_{11}$	Model 3
Level **2**	
$OCB=\gamma_{0}+\gamma_{21}\times GTL+\gamma_{22}\times CM+{}_{23}(GTL\times CM\;)+\mu_{21}$	Model 2
$PCT=\gamma_{1}+\gamma_{24}\times GTL+\mu_{22}$	Model 4
$OCB=\gamma_{2}+\gamma_{25}\times GTL+\gamma_{26}\times CM+\gamma_{27}\times PCT+\gamma_{28}(GTL\times CM\;)+\mu_{23}$	Model 5

Note: Mediation effect = $\gamma_{24}\times \beta_{11}$.

**Table 6 behavsci-13-00848-t006:** Results of aggregation test.

Variables	r_wg_	ICC (1)	ICC (2)
GTL	0.940	0.497	0.890
CM	0.831	0.531	0.915

Notes: GTL = group-focused transformational leadership; CM = collectivism.

**Table 7 behavsci-13-00848-t007:** Results of the cross-level regression analysis.

	OCB					PCT
	Model 1	Model 2	Model 3	Model 5	Model 6	Model 4
**Individual level**						
Intercept	3.165 ***	4.560 ***	3.280 ***	5.050 ***	3.721 ***	
Gender	0.042	0.044	0.013	0.034	0.032	
Age	0.003	−0.021	0.023	0.003	0.014	
Teaching experience	0.009	0.026	−0.003	0.021	0.006	
Educational background	−0.027	−0.018	−0.035	−0.028	0.043	
Professional title	−0.017	0.002	−0.046	−0.031	−0.011	
PCT			0.697 ***	0.608 ***	0.356 ***	
**Group level**						
GTL		0.881 ***		0.414 ***	0.347 ***	0.876 ***
GTL × CM					0.357 ***	

Notes: GTL = group-focused transformational leadership; OCB = organizational citizenship behavior of teachers; PCT = psychological contract of teachers; *** represents significance level *p* < 0.001; Model 3 is an individual-level analysis; Model 2, Model 4, Model 5, and Model 6 are a cross-level analysis.

## Data Availability

The datasets generated during and/or analyzed during the current study are available as a form of Appendix A and/or from the corresponding author upon reasonable request.

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
