# Peer review of "Cross-Level Influence of Group-Focused Transformational Leadership on Organizational Citizenship Behavior among Chinese Secondary School Teachers"

_behavsci, 2023, doi:10.3390/bs13100848_

Round 1
Reviewer 1 Report
Introduction
1. Excellent work on the first and second paragraphs.
2. For the third paragraph discussing the Chinese context, consider relocating this information to the methodology section, framing it as research context.
3. It seems that the content from p.2 line 85 to p.3 line 133 primarily reviews transformational leadership literature in relation to organizational citizenship behavior. This might fit better in the literature review section. Try summarizing the research niches of transformational leadership in a single paragraph for greater clarity.
4. What is the research question or statement? It needs to be clearly stated.
Literature Review
1. If Self-determination theory serves as the framework for your study, introduce it in the introduction. Then, link it to the research niches, explaining how it helps fill the research gaps.
2. The argument in section 2.1 seems reasonable, but it needs a clearer hypothesis. Consider integrating the argument from 2.1 into sections 2.2, 2.3, and 2.4. Then, present the theoretical framework figure at the end to avoid concept duplication. This is just a suggestion.
3. While the argument is sound, I have a question. Self-determination theory emphasizes both intrinsic and extrinsic motivation. However, it seems that the psychological contract, according to your framework, is influenced by external factors, possibly overlooking individual beliefs. Could you address this concern?
Methodology
1. Please specify the sample items for each construct.
2. The data analysis section appears to be missing. I recommend moving the reliability and validity sections to this part. Moreover, the standard procedure should be explained. Will you state the equation you used?
Results
1. The three-way interaction figure is clear.
Discussion
1. The theoretical contribution section needs a revision. It currently repeats what you've discussed in previous sections. The author might consider emphasizing how extrinsic factors influence or interact with intrinsic factors. This could be a potential direction.
Author Response
Thank you for your comments, please see the attachment.

Reviewer 2 Report
Dear authors,
Thank you for submitting this manuscript. The title of the paper is interesting. It invites readers to become curios to find out how organizational citizenship behavior can influence the organizational effectiveness and even become a competitive advantage for secondary schools in China. The aim of the paper is to reveal the mediating role of individual-level psychological contract fulfillment and the moderating impact of group-level collectivism.
Lines 10-25: The Abstract provides a pertinent overview of the work. The abstract relates to the title and the content of the paper. The purpose of the paper is clearly expressed: ”cross-level influence of group-focused transformational leadership on organizational citizenship behavior among secondary school teachers” and so are the methodology and findings.
Lines 29 - 164: The Introduction is long and descriptive. The current state of the research field is carefully reviewed, and significant publications cited. However, it does not show the importance of the study and how the finding could be of benefit and for what/whom.
Suggestions: Please briefly mention the main aim of the work and highlight the main conclusions in a final paragraph of the Introduction sections.
Lines 167 – 276: The theoretical framework has a good quality of the presentation. The literature review is appropriate. There is still room for further explanations regarding the main concept – of transformational leadership. This concept is explained from different elements – the relationship with the subordinates' basic psychological needs, how it influences influences subordinates' organizational citizenship behavior, mechanisms etc. What transformational leadership is does not appear very clear. In lines 234-246 this concept appears as focused on either individual or group level.
Suggestions: clarify the concept of transformational leadership from the very beginning, as what it really is first and then based on how it should be characterized. For example, transformational leadership should be considered as a relationship approach, or style of management, or a management philosophy, or an organizational process, or all together (a diagram would be nice to reveal such an understanding).
Lines 279 – 420: The 3 hypotheses are clearly formulated, based on the research framework, looking to identify if group-focused transformational leadership significantly and positively influences teachers' organizational citizenship behavior, if psychological contract fulfillment plays a mediating role between group-focused 385 transformational leadership and teachers' organizational citizenship behavior and in the end, if collectivism significantly and positively moderates the cross-level relationship between group-focused transformational leadership and individual-level teacher organizational citizenship behavior. Congratulations!
Lines 422 - 466: The selection of schools and respondents are clearly good explained. Few more clarifications would be important to be added: 1. why any 15 teachers in any teaching and research group were asked to fill in the questionnaires? Why not 15, or 20, or a proportion considering the size of the school?; 2. How were the questionnaires filed in? On paper? It is just an assumption that it was on paper and not electronically filled in. 3. …”the personnel departments of each school were responsible for collecting the questionnaires and mailing them back to the investigators”. How did they become responsible? Was any incentive provided? Who decided the responsibilities? 4. Was any pre-testing for the questionnaire? Why no/How? 5. Who paid for the mailing costs?
Suggestions: Please, address the abovementioned questions.
Lines 467 – 518: The measurement tools are statistically adequately used, with reliability demonstrated.
Lines 519 – 671: Findings are clear on the statistical software used and the results obtained. The interpretation of the results are significant, including the hypotheses testing.
Lines 672 – 784: Discussion and conclusion section is extensive. It also contains a subsection for theoretical implications. Are these the only implications? If yes, these should be integrated in section 5. If not, a subsection on practical implications should be considered.
Suggestions: Please, address the abovementioned comment.
Lines 826 – 1072: According to the authors’ instructions, “References must be numbered in order of appearance in the text (including table captions and figure legends) and listed individually at the end of the manuscript” (https://www.mdpi.com/journal/behavsci/instructions).
Suggestion: Please re-write the reference list according to the journal’s requirements.
Author Response
Thank you for reviewing our manuscript, please see the attachment.

Author Response

(The authors gave the same response as above.)

Round 2
Author Response
Dear reviewer,
I hope this letter finds you well. I am writing to provide an update on the status of my manuscript titled " Cross-Level Influence of Group-Focused Transformational Leadership on Organizational Citizenship Behavior Among Chinese Secondary School Teachers".
The referee's suggestions are incorporated in the revised manuscript. The following is our response to each comment in detail. Referee comments appear in italics; our responses are in normal typeface. Language in the manuscript has been proofread by MDPI English Editing team, and the revised portions have been highlighted for your convenience.
Comment 1: This study aims to examine the impact of Group-focused transformational leadership on an individual-level outcome variable known as Organizational Citizenship Behavior (OCB). It seeks to provide a more comprehensive explanation of this relationship through the inclusion of individual-level mediator variable, namely, the psychological contract, and a group-level moderator variable, collectivism. However, it appears that the explanation of the group-level moderator variable, collectivism, is overlooked in '2.1. Theoretical Framework,' and such content is omitted in the summarization on page 6, 2nd paragraph. It seems necessary to enhance the explanation of Hypothesis 3 in 'section 2.1.'
Response 1: Thank you for your valuable suggestions, in response to your comment regarding the omission of an explanation for the group-level moderator variable, collectivism, we have made substantial additions to the '2.1 Theoretical Framework' section of the manuscript. Specifically, we have included a thorough explanation of the concept of collectivism and how it relates to our research framework.
In the revised '2.1 Theoretical Framework' section, we have introduced the basic concept of collectivism and further elaborated on its connection to our research framework through the concept of leader-follower value congruence. This addition enhances the comprehensiveness of our theoretical foundation and provides readers with a better understanding of how collectivism fits into our study.
Comment 2: Please unify the variable names mentioned below.
- Variable names of the research model of figure 1
- Variable names in Hypothesis 1-3
- Variable names of '3.2. Measurement Tools'
- Variable names in Table 3-6
- Variable names in Table 2
- psychological contract of teacher vs. psychological contract
- teacher organizational citizenship behavior vs. organizational citizenship behavior of
teachers vs. Teachers' Organizational Citizenship Behavior
Response 2: We fully understand the importance of consistency in variable names to enhance the clarity and readability of the paper. In response to your valuable feedback, we have made the necessary revisions to unify variable names in various sections of the manuscript as 'psychological contract of teachers' and 'organizational citizenship behavior of teachers'."
Comment 3: In terms of academic English written, the paper needs to be proofed.
Thank you for your feedback regarding the need for proofreading the paper in terms of academic English writing. We appreciate your attention to detail and your commitment to enhancing the quality of our manuscript.
Response 3: Thank you, we have taken your suggestion to heart and enlisted the assistance of MDPI English Editing services to improve the quality of English expression throughout the manuscript. We believe that this professional editing service has greatly contributed to the overall clarity and readability of the manuscript. Also, the English editing certificate has been submitted along with this letter.
Comment 4: Throughout the entire article, several typos must be corrected. As an illustration:
- page 16) ‘the regression coefficient of group-focused transformational leadership on teacher organizational citizenship behavior ...’
- page 17) Figure 2. Collectivism as moderator of the relationship between GTL and OBC.
Response 4: We appreciate your careful review of our manuscript and for pointing out the instances of typos and errors throughout the manuscript. Your attention to detail is valuable in ensuring the overall quality of our work. We have thoroughly reviewed the manuscript and made the necessary corrections to address the typos you identified, including the examples you provided on pages 16 and 17.
Thank you once again for your careful review, and we remain open to any further suggestions or comments you may have.
Sincerely,
Zhuotao Fang
Shun-Chi Yu
